# Observational Study on the Impact of Large-Scale Photovoltaic Development in Deserts on Local Air Temperature and Humidity

**Wei Wu [1], Shengjuan Yue [1,2], Xiaode Zhou [1,*], Mengjing Guo [1], Jiawei Wang [1], Lei Ren [1] and Bo Yuan [1]**

[1]  State Key Laboratory of Eco-Hydraulics in Northwest Arid Region of China, Xi'an University of Technology, Xi'an 710048, China; wuwei@xaut.edu.cn (W.W.); yueshengjuan@163.com (S.Y.); guomengjing@xaut.edu.cn (M.G.); jiawei5066@hotmail.com (J.W.); ra91057@163.com (L.R.); yuanbo7327884@163.com (B.Y.)

[2]  State Key Laboratory of Plateau Ecology and Agriculture, Qinghai University, Xining 810016, China

*  Correspondence: zhouxd@mail.xaut.edu.cn; Tel.: +86-29-82312780; Fax: +86-29-83239907

**Abstract:** As an important form of clean energy, photovoltaic (PV) power generation is entering a rapid development phase. Qinghai, China is located on the Qinghai-Tibet Plateau. It has sufficient sunlight and rich heat and light resources, includes a large area of the Gobi Desert, and has become China's largest base for PV power generation. However, large-scale PV development in deserts changes the local surface energy distribution and impacts local microclimates. This study considered the Gonghe PV Power Plant in Qinghai as an example. Three monitoring stations were set up in the PV power plant, transition, and reference areas, and the influence of large-scale PV developments on the local air temperature and humidity was studied based on long-term, multi-point field observation data. The results showed that the overall daytime air temperature in the PV power plant had changed slightly (increased and decreased), while the night-time temperature dropped significantly. Specifically, in spring and summer, the daytime temperature increased slightly, with a maximum increase of 0.34 °C; in autumn and winter, the daytime temperature decreased slightly, with a maximum decrease of 0.26 °C; in all seasons, the night-time temperature decreased, with a maximum decrease of 1.82 °C during the winter night. The relative humidity in the PV power plant generally increased; except for a slight decrease in summer, the daytime and night-time relative humidity in spring, autumn, and winter always increased. The humidification in winter was the most significant, with increases of 5.00% and 4.76% for the transition and reference areas, respectively. The diurnal air temperature and relative humidity ranges in the PV power plant were greater than those outside the PV power plant. The results obtained in this field observation study could serve as a basis for quantitative evaluation of the microclimate effects of large-scale PV development in deserts and provide technical support for guiding the future planning and development of the PV industry.

**Keywords:** desert area; large-scale photovoltaic power plant; field observation; air temperature; relative humidity

## 1. Introduction

The global energy structure is currently undergoing profound changes. Traditional fossil-fuel-based energy development models are not sustainable. Clean and low-carbon renewable energy is the ultimate goal of global energy development [1]. Grossi [2] stated in the report of the United Nations Framework Convention on Climate Change (COP 25) Conference that despite the continuous investment in renewable energy, global emissions of greenhouse gases reached a record

high last year, so there is an urgent need to deploy various low-carbon resources, such as hydro, wind, and solar, as well as nuclear power and battery storage, to achieve climate goals. Solar energy is one of the fastest-growing renewable energy sources. Over 500 gigawatts (GW) of capacity was installed at the end of 2018, and by the end of 2019, there will most likely be a total global installed capacity of well over 600 GW [3]. Solar photovoltaic (PV) reduces $CO_2$ emissions by 2.3 giga-tonnes of per year [4]. The number of large-scale PV power plants is increasing rapidly worldwide [5]. According to a report by the International Energy Agency, the installed capacity of PV worldwide exceeded 400 GW at the end of 2017 [6]. China's cumulative installed PV capacity is 131 GW, accounting for 32.57% of the world's total cumulative installed PV capacity and ranked first [7]. China is the largest $CO_2$ emitting country in the world, which accounts for 28% of the $CO_2$ emissions globally [8]. Therefore, the reduction of carbon dioxide emissions from China's PV industry has a direct impact on global trends.

Qinghai, China is located in the Qinghai-Tibet Plateau. The sufficient sunlight, rich solar and thermal resources, and large area of the Gobi Desert represent natural advantages for the development of the PV industry, and Qinghai is currently China's largest base of PV power generation [9]. However, the Qinghai-Tibet Plateau is a typical fragile ecological region that is environmentally unique and sensitive. The construction and operation of large-scale PV power plants have changed the surface albedo and land cover [10]. PV panels change the surface energy balance and heat fluxes by absorbing radiation [11], which inevitably influences local climates and even the global climate [12]. In addition, the Qinghai-Tibet Plateau is one of the most fragile and typical ecological regions in the world, and its environmental specificity and sensitivity are significant. Therefore, the impact of constructing large-scale desert PV power plants on the local climate in such areas is even more noteworthy.

Researchers have obtained some research conclusions on the local climate impacts of PV power plants in specific regions based on numerical simulations and field observations. For example, the simulation results of a climate model for PV on urban roofs showed that during the summer, the temperature decreased during the day (0.28–0.48 °C) and at night (0.38–0.78 °C) [13]. Climate simulations of large-scale PV power plants in the Sahara showed an increase in temperature (maximum temperature (+1.28 K) and minimum temperature (+0.97 K)) [14]. Field observations of the Tucson PV Power Plant in the United States showed that it has a "heat island effect", with an average annual temperature increase of 2.4 °C at a height of 2.5 m and a temperature increase of 3-4 °C at night [15]. Field observations of the Red Rock PV Power Plant in the United States showed that the maximum daytime temperature increased by 1.38 °C at a height of 1.5 m, while there was no significant difference in the night-time temperature [16]. This is contrary to the conclusion of the Tucson PV Power Plant [15], possibly due to the smaller and less continuous PV array, and the advection of adjacent impervious areas in the PV power plants affects the temperature measurement. Field observations of the Golmud PV Power Plant in Qinghai, China showed that the temperature field at 2 m was higher than that of the off-site control area, with a maximum temperature difference of 0.67 °C, while the temperature at 10 m was lower [17]. Many related studies on the Gonghe PV Power Plant in Qinghai, China have been conducted. Yin et al. [18] analyzed field observation data for two months and found that in the PV power plant, the temperature at 2 m was higher in the daytime than at the control point, while the temperature at night was significantly lower. However, due to the short observation data sequence, no comprehensive and systematic conclusions on temperature change were obtained. Chang et al. [19] showed that the rising temperature of PV panels during the daytime is a heat source for the surrounding environment, which may cause the "heat island effect", while PV panels have a cooling effect at night. The different temperature changes in the above different studies may be due to areas with different land-surface (such as urban and desert) properties, which give diverse fluxes of sensible and latent heat, which in turn affect the turbulent mixing and horizontal advection in the atmospheric boundary layer [20]. Regarding the influence of PV power plants on air humidity, Yin et al. [18] showed that the relative humidity at 2 m height changed in the same way inside and outside the station during the day, but inside the station at night was significantly higher than outside the station. The increase in local humidity at PV power plants provides favorable conditions for vegetation growth in desert areas [21].

Additionally, increased humidity can increase the viscosity of the sand surface, so the sand particles are not easily driven by wind and sand drift activity, which reduces the frequency of sandstorms [22] and thereby also reduces the impact of dust on the efficiency of PV power plants [23]. Based on the above research, it can be seen that PV power plants have a significant impact on air temperature and humidity, which in turn will affect the surface temperature and regulate the ecological environmental climate. Therefore, the impact of large-scale PV power plants on the climate in desert areas is worth a comprehensive study.

Given these issues, this article took the Gonghe PV Power Plant in Qinghai as the study area and set up three monitoring stations in the power plant area, the transition area, and an off-site reference area. Based on long-term, multi-point field observation data from January 2019 to December 2019, the impact of a large-scale PV development in the desert on the local air temperature and relative humidity was studied, and the characteristics and variations in the air temperature and relative humidity were evaluated on different time scales. The results could lay the foundation for research on the impact of large-scale PV development in deserts on the local ecological environment.

## 2. Study Area and Method

### 2.1. Overview of the Study Area

The study area of this article is located in the Talatan PV power generation area in Gonghe County (hereafter referred to as the Gonghe PV Power Plant), Qinghai Province (Figure 1a). The area is 12 km away from Gonghe County. The local climate has prominent characteristics of a continental plateau. The annual average temperature is 4.1 °C, the annual precipitation is 246.3 mm, and the annual evaporation is 1716.7 mm. The annual average wind speed is 2.1–2.7 m/s, and the annual average number of windy days is 17.7–43.2 d, with a maximum of 75 d. The annual average number of sandstorm days is 6.5–20.7 d. The annual average solar radiation is 6564.26 MJ/m$^2$, and the annual average sunshine duration is 2907.8 h [24].

### 2.2. Study Method

The PV array in the Gonghe PV Power Plant is mainly composed of stationary PV panels. Its azimuth is due south, the inclination is 34°, and the distance between PV panels is 7.5 m. There were three monitoring sites in this study area (Figure 1b). The first monitoring site (hereinafter referred to as the PV power plant monitoring site (PV site), Figure 1c) was located in the PV array at 36.131° N and 100.567° E, and the altitude was approximately 2913 m. The second monitoring site (hereinafter referred to as the transition area monitoring site (TRA site), Figure 1d) was located in the transition area between the PV array and the desert. It was 6.65 km southwest of the PV site at 100.507° N and 36.096° E, and the altitude was approximately 2924 m. The third monitoring site (hereinafter referred to as the reference site (REF site), Figure 1e) was located in the desert outside the PV power plant. It was 8.17 km to the northwest of the PV site at 100.509° N and 36.1187° E, and the altitude was approximately 2922 m. The three monitoring stations used the same types of instruments. The air temperature and relative humidity sensors used were HMP155A models with a temperature accuracy of ±0.12 °C and a relative humidity accuracy of ± 1% (0–90% relative humidity). The air temperature and relative humidity instrument at each monitoring station was mounted 2.5 m above the ground. The data collectors were CR1000X synchronous collectors, recording every 30 minutes. Data from January 2019 to December 2019 from the PV, TRA, and REF sites were used in this study. The monthly average diurnal values were calculated as the arithmetic means of the values at corresponding times each day throughout the month.

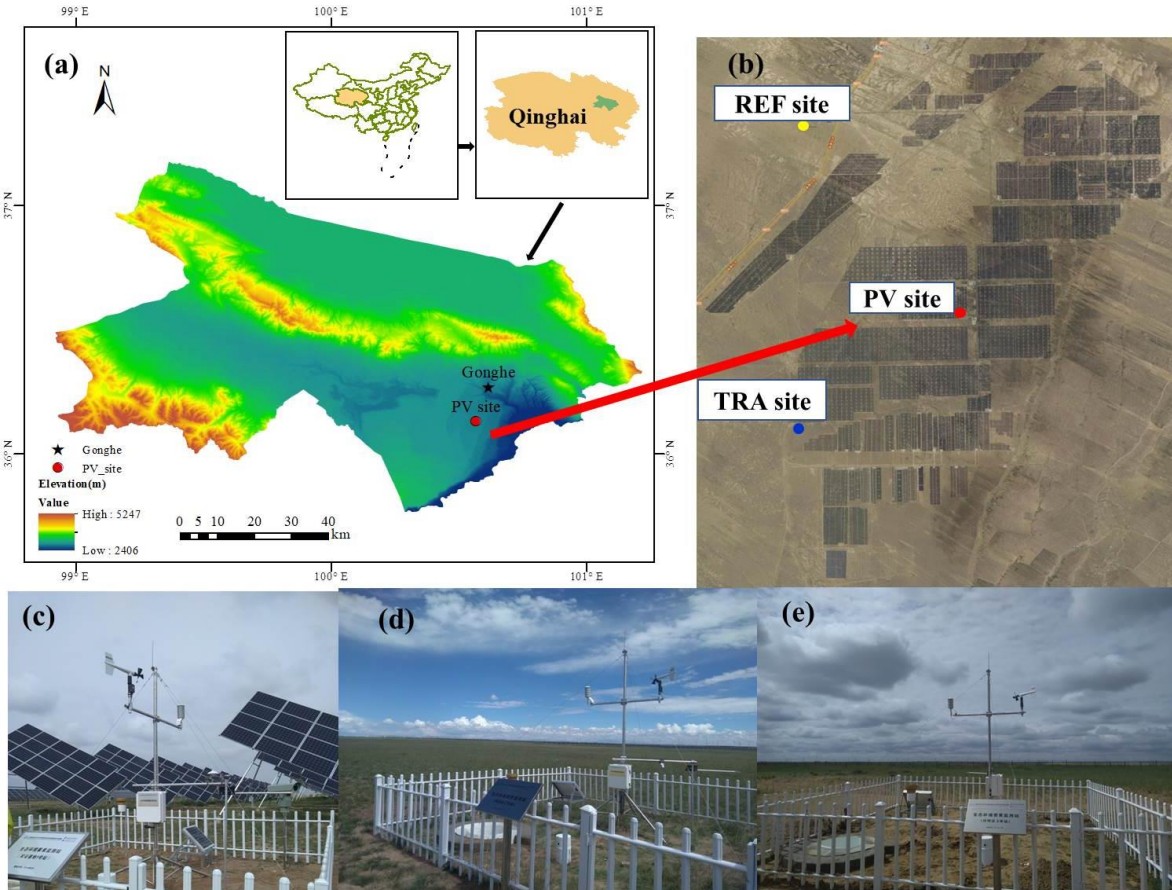

**Figure 1.** (**a**) The location of the Gonghe Photovoltaic (PV) Power Plant in the Qinghai province. (**b**) The relative position of the monitoring sites. (**c**) PV monitoring site. (**d**) Transition area (TRA) monitoring site. (**e**) Reference (REF) monitoring site.

## 3. Results and Analysis

### 3.1. Temperature Variations

#### 3.1.1. Diurnal Temperature Variations

Figure 2 shows the monthly average diurnal variations in 2.5 m air temperature at the PV, TRA, and REF sites. The temperature of the three monitoring sites was minimum at approximately 8:00 and maximum at approximately 16:30. Depending on the season, the maximum or minimum value occurred slightly earlier or later. The average annual temperatures at the PV, TRA, and REF sites were 3.54, 4.07, and 4.10 °C, respectively. The temperatures at the PV site were 0.53 and 0.56 °C lower than those at the TRA and REF sites, respectively. The temperatures at the three monitoring sites tended to be the same during the day (shaded part in Figure 2) but were significantly different at night.

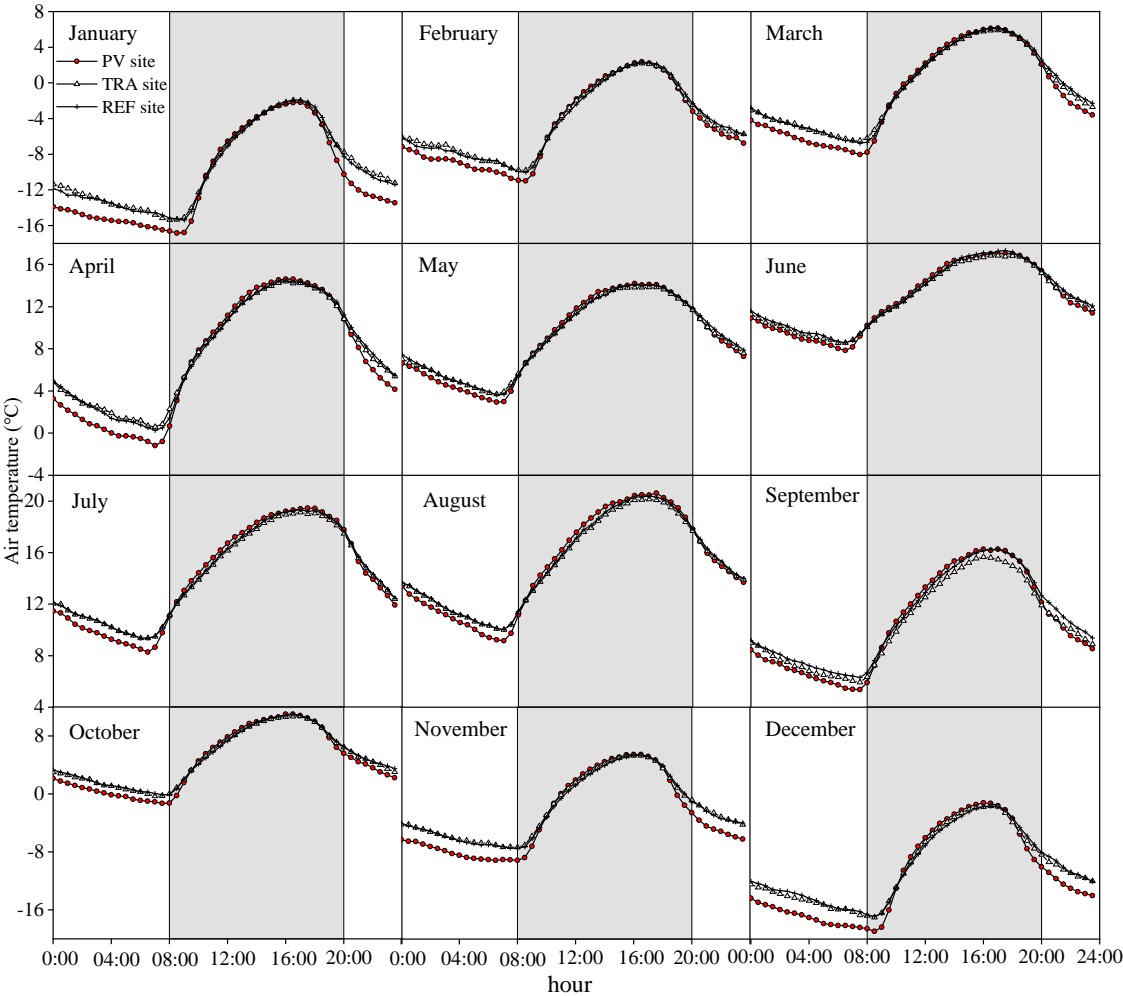

**Figure 2.** The monthly average diurnal variations in temperature (the shaded part of the figure represents the daytime).

### 3.1.2. Daytime and Night-Time Variations in Temperature Difference

Since the main working hours of the Gonghe PV Power Plant are during the day, there were significant differences in day and night temperatures. The temperatures were divided into daytime and night-time temperatures for discussion. The daytime temperature is the average of temperatures measured between 08:00 and 20:00 local time. The night-time temperature is the average of temperatures measured between 21:00 and 07:00 local time. The average diurnal temperature is the average of temperatures measured during the same day (24 h). Figure 3 compares the variations in the difference between the average daytime, night-time, and diurnal temperatures at the PV, TRA, and REF sites. The PV power plant had a warming effect during the day in spring and summer. The temperature difference between different monitoring sites was larger in summer than in spring, and the average daytime temperature at the PV site in summer was 0.34 and 0.20 °C higher than those at the TRA and REF sites, respectively. This greater impact of the PV power plant on the ambient temperature in summer might be due to the higher solar radiation intensity in the desert and the longer operating time of the PV power plant in summer. The PV power plant had a cooling effect during the day in autumn and winter, and the cooling effect was more prominent in winter than in autumn. The average daytime temperature at the PV site was 0.26 and 0.22 °C lower than those at the TRA and REF sites, respectively, in winter.

The night-time temperature at the PV site was always lower than those at the TRA and REF sites, so the PV power plant had a cooling effect at night. Furthermore, the closer the monitoring site was to

the center of the PV power plant, the smaller the night-time temperature drop was at the monitoring site. The night-time cooling effect of the PV power plant was weakest in summer, and the average night-time temperatures at the PV site in summer were 0.51 and 0.59 °C lower than those at the TRA and REF sites, respectively. This weaker effect mainly occurred because the daytime warming effect in summer reduced the magnitude of night-time cooling. The night-time cooling effect of the PV power plant was strongest in winter, and the average night-time temperatures at the PV site in winter were 1.81 and 1.82 °C lower than those at the TRA and REF sites, respectively.

As the daytime warming effect was weaker than the night-time cooling effect at the PV site, the PV power plant showed an overall cooling effect throughout the day. This result was inconsistent with the conclusions of Chang et al. [19] from a study of the same PV power plant. This difference might be due to the selection of different monitoring sites. Chang et al. [19] used the temperature of PV panels to represent the temperature of the PV power plant. The daytime increase in PV panel temperature was significantly greater than the night-time decrease in PV panel temperature. In spring and summer, the temperature difference between the PV and TRA sites was smaller than the temperature difference between the PV and REF sites. This result indicated that in the warm season, the closer the monitoring site was to the center of the PV power plant, the smaller the temperature difference was between the two locations. In autumn, the temperature differences between the PV and TRA sites and between the PV and REF were similar. In winter, the temperature difference between the PV and TRA sites was greater than that between the PV and REF. This result indicated that in the cold season, the closer the monitoring site was to the center of the PV power plant, the greater the temperature difference was between the two locations.

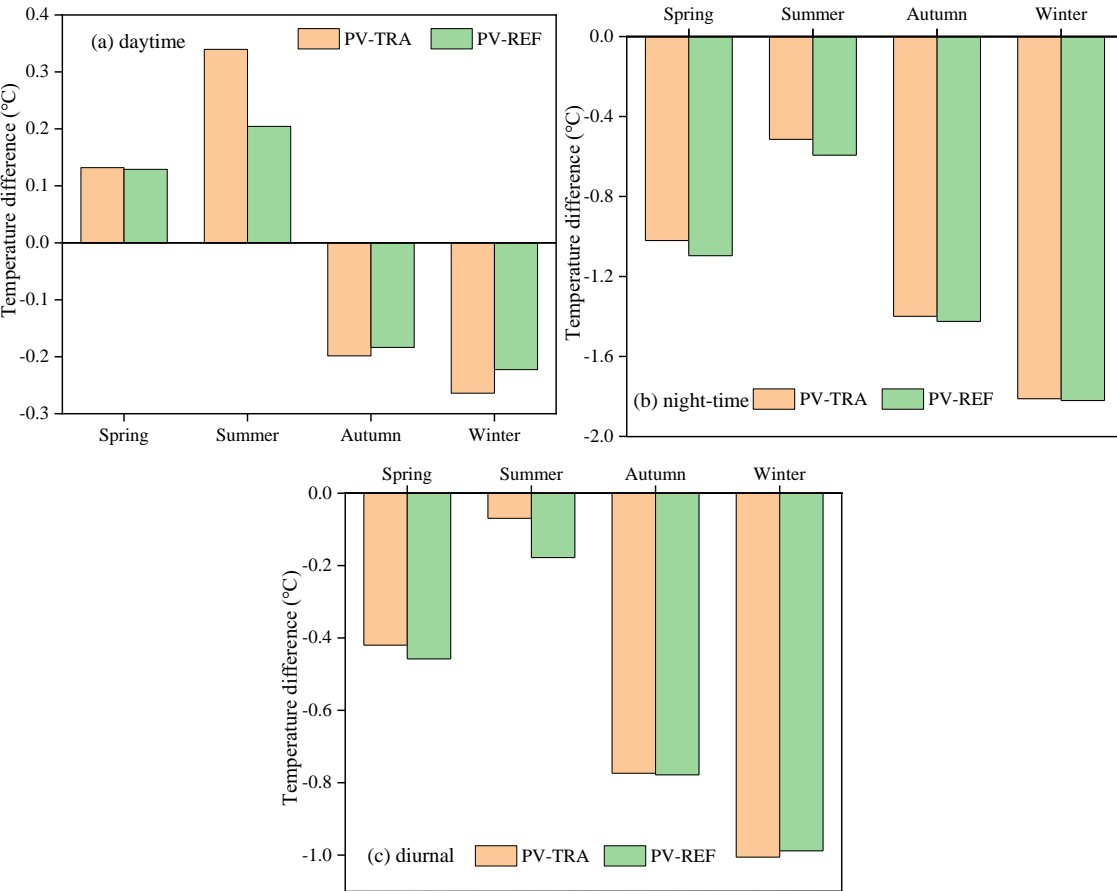

**Figure 3.** The variations in the difference between the average daytime, night-time, and diurnal temperatures: (**a**) Daytime, (**b**) night-time, and (**c**) diurnal.

### 3.1.3. Variations in the Diurnal Temperature Range

Compared with variations in average temperature, the diurnal temperature range can more effectively reflect the impact of a PV power plant on local temperature. Table 1 shows the diurnal temperature range at the PV, TRA, and REF sites. As the PV power plant area exhibited a warming effect during the day and a cooling effect at night, the diurnal temperature range at the PV site was significantly greater than those at the TRA and REF sites, while the diurnal temperature ranges at the TRA and REF sites (outside the power plant) were similar. The maximum and minimum diurnal temperature ranges at the PV site occurred in December and June, respectively. The diurnal temperature range at the PV site had a peak in April mainly because during the spring–summer transition period, the maximum temperature in the study area increased, while the minimum temperature was still below 0 °C, making the diurnal temperature range large. For the transition area, the impact of the PV power plant on the diurnal temperature range was greatest (19.11%) in September. For the reference area, the impact of the PV power plant on the diurnal temperature range was delayed and was greatest (11.66%) in October. This result showed that with increasing distance of the monitoring site from the center of the PV power plant, the influence of the PV power plant on the diurnal temperature range at the monitoring site gradually decreased.

**Table 1.** The diurnal temperate ranges at the PV, TRA, and REF sites (unit: °C). Bold data represent the greatest value.

| Month | 2019/1 | 2 | 3 | 4 | 5 | 6 | 7 | 8 | 9 | 10 | 11 | 12 |
|---|---|---|---|---|---|---|---|---|---|---|---|---|
| PV site | 17.36 | 15.76 | 15.63 | 18.46 | 14.19 | 11.60 | 13.57 | 13.17 | 12.92 | 14.31 | 17.37 | **19.72** |
| TZ site | 15.52 | 13.90 | 14.65 | 16.75 | 12.90 | 10.27 | 11.60 | 11.40 | 10.45 | 12.76 | 14.78 | **17.51** |
| REF site | 15.91 | 14.54 | 15.37 | 17.05 | 13.01 | 10.60 | 12.02 | 12.06 | 11.64 | 12.64 | 15.66 | **17.56** |
| PV-TZ/PV (%) | 10.59 | 11.81 | 6.32 | 9.27 | 9.13 | 11.51 | 14.50 | 13.47 | **19.11** | 10.80 | 14.90 | 11.19 |
| PV-REF/PV (%) | 8.33 | 7.76 | 1.72 | 7.66 | 8.35 | 8.65 | 11.37 | 8.48 | 9.93 | **11.66** | 9.81 | 10.95 |

### 3.2. Variations in Relative Humidity

### 3.2.1. Diurnal Variations in Relative Humidity

The monthly average diurnal variations in relative humidity at 2.5 m at the PV, TRA, and REF sites are shown in Figure 4. The monthly average diurnal change trend of relative humidity was exactly the opposite of the air temperature trend. When the temperature was the highest at around 16:30 (the lowest at around 8:00), the relative humidity was the lowest (highest). The relative humidity also showed a small difference during the day but a significant difference at night. This is consistent with the conclusion of the relative humidity change in the Golmud PV power plant [17]. The average annual relative humidities at the PV, TRA, and REF sites were 57.66%, 55.23%, and 54.66%, respectively. Compared with the TRA and REF sites (outside the PV power plant), the humidity inside the PV power plant was higher by 2.43% and 3.00%, respectively.

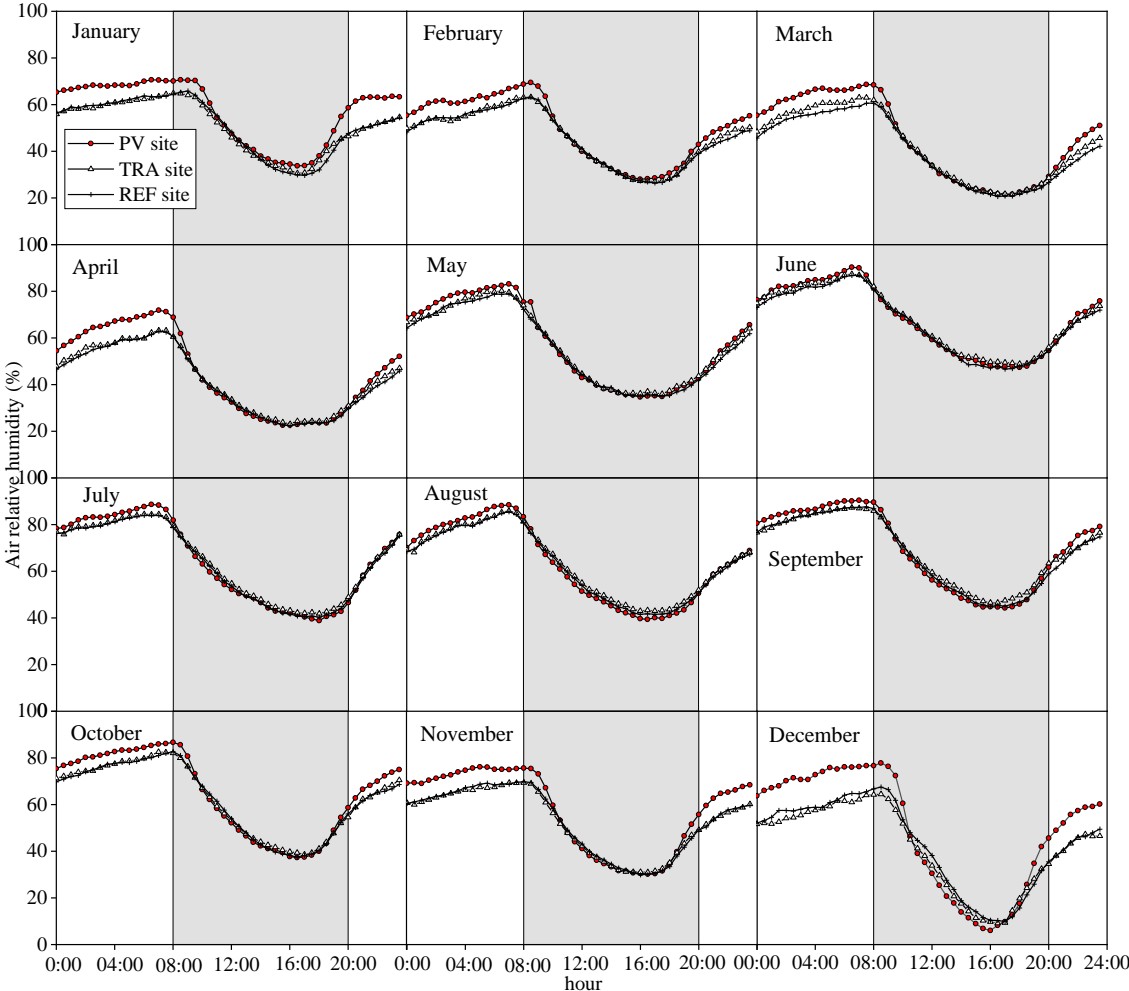

**Figure 4.** The monthly average diurnal variations in relative humidity (the shaded part of the figure represents the daytime).

### 3.2.2. Daytime and Night-Time Variations in Relative Humidity Difference

The comparison of the monthly average diurnal variation in the difference in relative humidity inside and outside the PV power plant is shown in Figure 5. When the daytime temperature was high, the relative humidity at the PV site was lower than those at the TRA and REF sites (except in January), indicating that the PV power plant had a dehumidification effect, but the dehumidification range is small. At night, the relative humidity at the PV site was significantly higher than those at the TRA and REF sites, indicating that the PV power plant had a significant humidification effect.

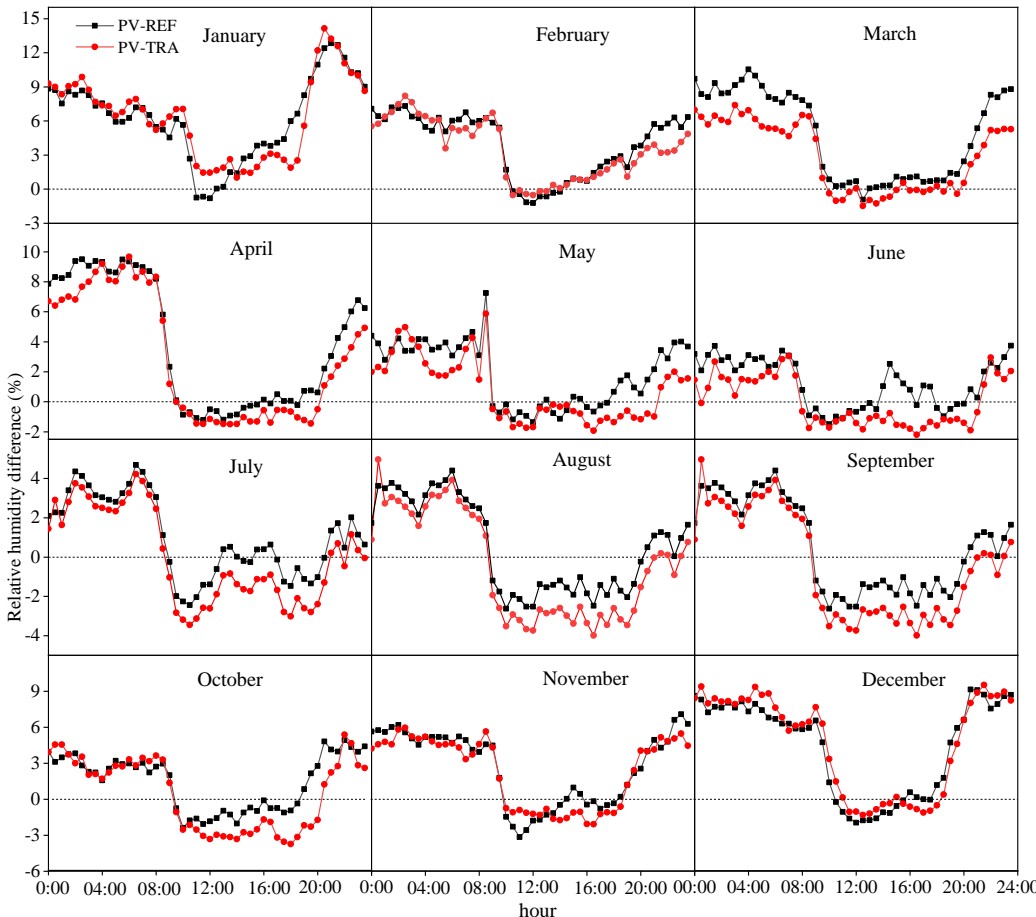

**Figure 5.** The monthly average diurnal variations in relative humidity difference.

The comparison of the seasonal variation in the difference in relative humidity inside and outside the PV power plant is shown in Figure 6. The PV power plant had a dehumidification effect during the day in summer, and the daytime relative humidity at the PV site was 1.91% and 0.75% lower than those at the TRA and REF sites, respectively. This result indicated that the closer the monitoring site was to the center of the PV power plant, the more significant the relative humidity decrease was at the monitoring site during the day in summer. In autumn and winter, the PV power plant had a humidification effect during the day. The daytime humidification effect at the PV site was more significant in winter than in autumn, and the daytime relative humidity at the PV site in winter was 2.48% and 2.24% higher than those at the TRA and REF sites, respectively. In spring, the daytime relative humidity at the PV site was lower than that at the TRA site and higher than that at the REF site. At night, the PV power plant had a humidification effect in all seasons. The humidification effect at night at the PV site was smallest in summer, and the night-time relative humidity at the PV site in summer was 1.77% and 2.57% higher than those at the TRA and REF sites, respectively. The humidification effect at night at the PV site was largest in winter; the night-time relative humidity at the PV site in winter was 7.75% and 7.50% higher than those at the TRA and REF sites, respectively. In general, the PV power plant had a humidification effect in all seasons (except for summer, during which the relative humidity at the PV site was 0.15% lower than that at the TRA site). The humidification effect of the PV power plant was the most significant in winter, and the relative humidity at the PV site was 5.00% and 4.76% higher than those at the TRA and REF sites, respectively.

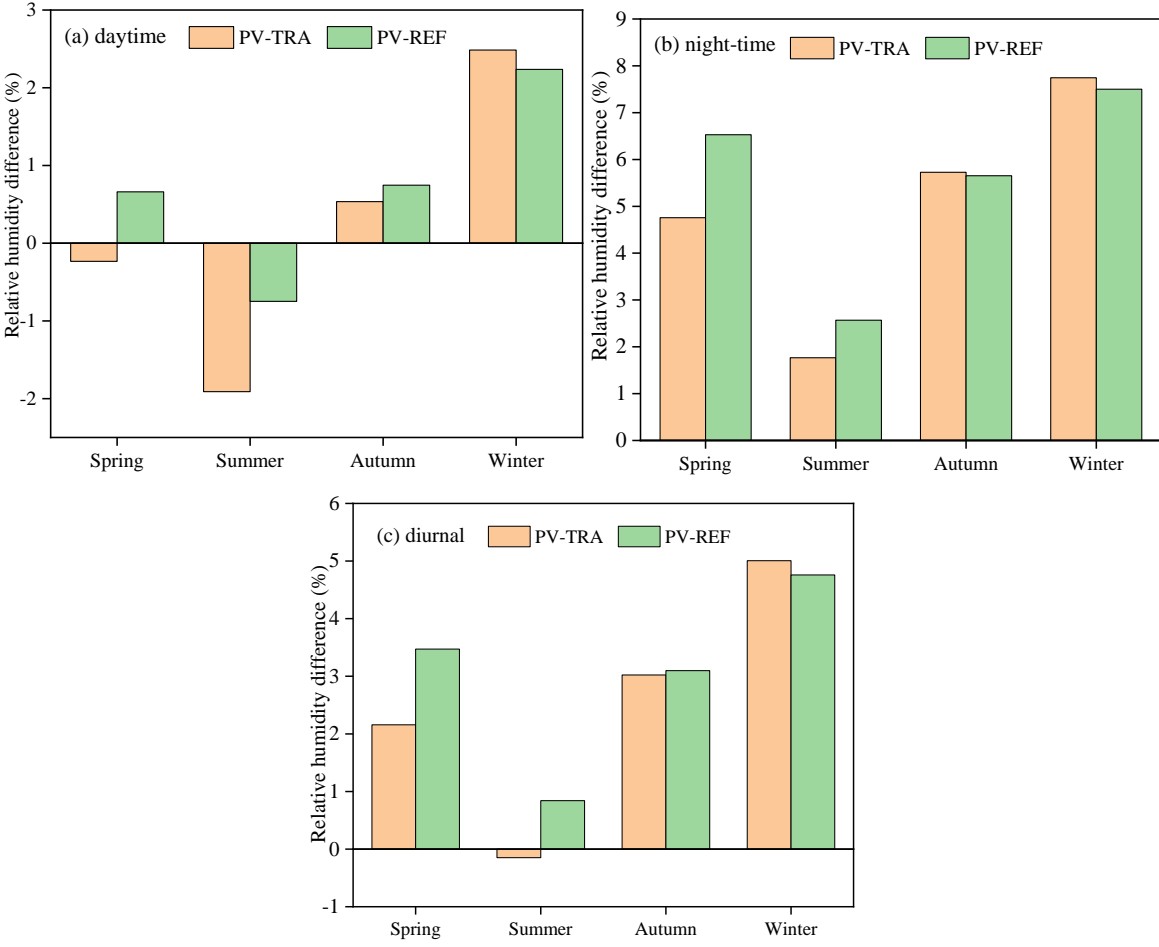

**Figure 6.** The variations in the differences between the average daytime, night-time, and diurnal relative humidities: (**a**) Daytime, (**b**) night-time, and (**c**) diurnal.

### 3.2.3. Variations in the Diurnal Relative Humidity Ranges

Table 2 shows the diurnal ranges of relative humidity at the PV, TRA, and REF sites. The patterns of the diurnal ranges of relative humidity at the three monitoring sites were the same; in other words, the diurnal ranges of relative humidity at the three monitoring sites increased or decreased at the same time. Furthermore, the diurnal range of relative humidity at the PV site was greater than those at the TRA and REF sites. This result indicated that the PV power plant had a significant impact on the relative humidity in the local area. The maximum diurnal range of relative humidity occurred in May, and the diurnal ranges of relative humidity at the PV, TRA, and REF sites in May were 65.61%, 60.62%, and 59.52%, respectively. This maximum was mainly due to the gradual increase in precipitation and gradually increasing evapotranspiration from vegetation that started to grow [25] in the study area in May. The minimum diurnal range of relative humidity at the PV site appeared in January, with a value of 50.17%. The minimum diurnal ranges of relative humidity at the TRA and REF sites were 42.72% and 46.50%, respectively, which appeared in December. For the transition area, the impact of the PV power plant on the diurnal range of relative humidity was the greatest (19.13%) in December. For the reference area, the impact of the PV power plant on the diurnal range of relative humidity was the greatest (12.08%) in April. These results showed that the influence of the PV power plant on the diurnal humidity range of a given monitoring site gradually decreased as the distance of the monitoring site from the center of the PV power plant increased.

**Table 2.** The diurnal relative humidity range at the PV, TRA, and REF sites (unit: %). Bold data represent the greatest value.

| Month | 2019/1 | 2 | 3 | 4 | 5 | 6 | 7 | 8 | 9 | 10 | 11 | 12 |
|---|---|---|---|---|---|---|---|---|---|---|---|---|
| PV site | 50.17 | 51.33 | 58.81 | 62.02 | **65.61** | 55.46 | 60.80 | 57.32 | 57.66 | 59.65 | 56.44 | 52.83 |
| TZ site | 44.69 | 47.30 | 53.73 | 55.42 | **60.62** | 50.42 | 57.00 | 54.00 | 49.28 | 54.86 | 48.73 | 42.72 |
| REF site | 46.84 | 47.23 | 52.81 | 54.53 | **59.52** | 52.05 | 58.27 | 55.23 | 54.38 | 56.99 | 52.49 | 46.50 |
| PV-TZ/PV (%) | 10.92 | 7.85 | 8.64 | 10.63 | 7.60 | 9.08 | 6.25 | 5.79 | 14.53 | 8.03 | 13.66 | **19.13** |
| PV-REF/PV (%) | 6.63 | 7.99 | 10.21 | **12.08** | 9.28 | 6.14 | 4.16 | 3.65 | 5.69 | 4.47 | 7.01 | 11.98 |

## 4. Discussion

### 4.1. Impact of the PV Power Plant on Local Temperature

In this study, the Gonghe PV Power Plant had a warming effect in spring and summer in terms of the 2.5 m daytime temperature (Figure 3a). The maximum temperature increase in summer was 0.34 °C, and the closer the monitoring site was to the center of the PV power plant, the greater the temperature increase was. This greater impact of the PV panels on the ambient temperature in summer might be due to the higher solar radiation intensity in the desert and the longer operating time of the PV power plant in summer. Daytime warming effects have also been reported in field monitoring studies of PV power plants in other regions in China and other countries. For example, the Golmud PV Power Plant had the largest daytime increase in 2 m temperature in the summer, with a value of 0.67 °C [26]. The locations of the temperature monitoring instruments in different studies affected the degrees to which PV power plants influenced temperature. For example, the daytime temperature of PV panels was up to 9.7 °C higher than the ambient temperature at a height of 2 m [19]. As the height from the ground increased, the heating effect of the PV power plant decreased, and the heat generated by the panels would be completely dissipated to the environment at a height of 5–18 m. As the distance of the monitoring site from the PV power plant increased, the thermal energy dissipated more quickly, and the air temperature approached the ambient temperature [27]. The PV power plant had a cooling effect in autumn and winter in terms of the 2.5 m daytime temperature (Figure 3a), and the maximum temperature reduction in winter was 0.26 °C. This effect might be due to the low solar radiation intensity in the cold seasons; the PV power plant absorbed part of the solar radiation and converted it into electrical energy; thus, the energy reaching the underlying surface inside the PV power plant was lower than that outside the plant [17].

In this study, the PV power plant had a cooling effect at night in all seasons (Figure 3b). The temperature drop was minimum in summer and maximum (1.82 °C) in winter. The conclusions of the impact of PV power plants on night-time temperature varied with different underlying surfaces. The 2.5 m night-time temperature at the desert-based Tucson PV Power Plant was 3 to 4 °C higher than that of the surrounding area, and the PV heat island effect in the warm seasons (spring and summer) was more significant than was the urban heat island effect [15]. The desert Golmud PV Power Plant also had a warming effect in terms of the 2 m night-time temperature, and the PV panels had an insulation effect on the near-surface layer [17]. In this study, the underlying surface of the Gonghe PV Power Plant was desert with sparse vegetation. Chang et al. [19] showed that the PV panels had a cooling effect at night, which is consistent with the conclusions of this article. Based on different climatic conditions, seasons, underlying surfaces, and scales of the PV power plants, the environmental temperature inside and outside PV power plants may decrease to different degrees at night.

In this study, the temperature in the PV power plant increased slightly during the day and decreased significantly at night. The PV power plant showed a cooling effect in terms of diurnal temperature variations during all four seasons (Figure 3c). The annual average temperature at the PV site was lower than those at the TRA and REF sites (outside the plant) by 0.53 and 0.56 °C, respectively. The PV power plant did not produce a heat island effect or potentially unfavorable microclimate [27]. This result might be related to the fact that the monitoring site was close to a road inside the PV power

plant, which had a spatial cooling effect [27]. In addition, advection on impervious roads may affect the monitored air temperature and humidity [16].

The diurnal temperature range of the PV power plant was significantly larger than those of the reference and transition areas. This difference reflected the significant impact of the PV power plant on local temperature. Diurnal variations in temperature have a direct or indirect impact on vegetation and soil processes [28]. For example, root respiration may adapt to the lowest temperature at night [29]. Additionally, the diurnal temperature range may affect crop yields [30]. For example, a large diurnal temperature range during the growing period of alfalfa can increase the vitality of its root system and increase the sugar content in the plant, which is beneficial to the accumulation of dry matter. These processes have a great impact on the yield and quality of alfalfa and are also very beneficial for its growth [31].

*4.2. Impact of the PV Power Plant on Local Humidity*

Large-scale PV power plants are mostly built in the Gobi Desert and desert areas with abundant solar energy resources and dry climates [32]. The water-holding capacity of the soil is low, the ecological environment is relatively fragile, and the humidity is sensitive to changes [33]. In this study, the Gonghe PV Power Plant had a dehumidification effect during the day in summer and a humidification effect during the day in winter, which might be caused by the unstable stratification of the near-surface atmosphere inside and outside the PV power plant during the day [13]. The daytime humidification effect was most prominent in winter (Figure 6a). The PV power plant had a night-time humidification effect in all seasons; this effect was smallest in summer and largest in winter (Figure 6b). This humidification effect may have occurred because the PV power plant made the near-surface air temperature at the PV site lower than that at the TRA and REF sites at night, thereby enhancing the temperature inversion of the boundary layer, which is not conducive to the upward transport of water vapor [14]. The PV power plant had a daytime humidifying effect in all seasons (except for summer, during which the humidity at the PV site was 0.15% lower than that at the TRA site, Figure 6c). There was no difference in the 2 m humidity inside and outside the Golmud PV Power Plant in the Gobi Desert [17]. The inconsistency in the differences in relative humidity inside and outside the two PV power plants might be caused by differences in atmospheric structure in the regions where the plants were located [34].

As PV panels convert some solar energy into electricity, the shielding provided by the solar panels reduces the amount of solar radiation absorbed by the ground surface and reduces the land surface temperature. As a result, the intensity of atmospheric heating of the land surface weakens, the evaporation rate of soil moisture decreases, a certain amount of precipitation stays longer after reaching the ground, and the humidity slightly increases [17]. Therefore, PV power plants play a role in humidifying in the local microclimate.

## 5. Conclusions

This study compared the characteristics of the 2.5 m air temperature and relative humidity at the PV, TRA, and REF sites of a large-scale PV power plant in a desert area and obtained the following conclusions. (1) The overall daytime temperature in the PV power plant had changed slightly (increased and decreased), while the night-time temperature dropped significantly. Specifically, in spring and summer, there was a slight warming trend in the daytime; in autumn and winter, there was a slight cooling trend in the daytime; in all seasons, the night-time temperature decreased. (2) The relative humidity in the PV power plant generally increased. (3) The diurnal temperature and relative humidity ranges in the PV power plant were always greater than those outside the plant. In the future, more field measurements at different heights should be observed and used to calibrate the results of climate models and satellite observations.

**Author Contributions:** M.G., J.W., L.R., and B.Y. collected the data. W.W. and S.Y. assessed the data and wrote the manuscript. X.Z. reviewed and edited the manuscript. All authors have read and agreed to the published version of the manuscript.

**Funding:** This work is supported by the National Natural Science Foundation of China (Grant No. 51979222 and 91747206).

**Conflicts of Interest:** The authors declare no conflict of interest.

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
