# Peer review of "Observational Study on the Impact of Large-Scale Photovoltaic Development in Deserts on Local Air Temperature and Humidity"

_sustainability, doi:10.3390/su12083403_

Round 1

Reviewer 1 Report

This paper studies the impact of large-scale PV development in deserts on local air temperature and humidity by observation. The data are well presented, and analysis is given. Besides, some potential reasons and insight of the temperature and humidity change could be given to make this paper more valuable.

Some other issues may help to improve the quality of this paper are listed.

Reference is required for P2 Line 55.

There are reference lumps in the P3 Line 69. Please eliminate this lump. After that please check the manuscript thoroughly and eliminate all the lumps. This should be done by characterising each reference individually.

Improve the quality of Fig. 2 and 4 to make them clearer when zoom in. Also, why there is are dark areas in these two figures.

According to the content in the literature review, it seems the temperature changes in different studies are varied, a comparison could be made to show some possible insight reasons.

If it is possible, it is better to show some data of these three monitoring sites before the PV power plant is constructed to eliminate the influence brought by different locations of monitoring sites.

Author Response

Response to Reviewer 1 Comments

Point 1: Reference is required for P2 Line 55.

Response 1: We have added a reference here, which is “International Energy Agency. 2018 World Energy Outlook: Executive Summary; Paris, 2018. ” (Line 63-64, Page 2)

Point 2: There are reference lumps in the P3 Line 69. Please eliminate this lump. After that please check the manuscript thoroughly and eliminate all the lumps. This should be done by characterising each reference individually.

Response 2: We have eliminated all the lumps in the manuscript.

Point 3: Improve the quality of Fig. 2 and 4 to make them clearer when zoom in. Also, why there is are dark areas in these two figures.

Response 3: Thank you for your suggestion. We have adjusted Fig. 2 and 4. The dark areas in the two figures represent the time of day, during which the air temperature and humidity of the three monitoring sites have very little difference.

Point 4: According to the content in the literature review, it seems the temperature changes in different studies are varied, a comparison could be made to show some possible insight reasons.

Response 4: Thank you for your suggestion. We have added reasons for the different temperatures in different studies.  (Line 87- 90, Page 3; Line 99- 103, Page 3)

Point 5: If it is possible, it is better to show some data of these three monitoring sites before the PV power plant is constructed to eliminate the influence brought by different locations of monitoring sites.

Response 5: Thank you for your suggestion. It would be a good suggestion to compare the influence of PV power plant before and after construction, but unfortunately, we don’t collect the data before the construction of the PV power plant.

The authors thank you very much again for your constructive comments which play a significant role to increase the quality of this paper.

Reviewer 2 Report

The scientific quality of the paper is very good, the topic is relevant to the journal Sustainability. However, I have some comments and questions about the article.

1.Line 125 - Fig. 1 Misspellings: The lactation of Gonghe PV Power Plant in Qinghai province

2. The data and descriptions in all graphs should be written in a larger font. In this form it is quite difficult to read them.

3. Line 203-204 Fig. 4 - Authors should use different colors or patterns to differentiate between curves. It is difficult to distinguish them in this form.

4. The paper is well written, but it still needs some english improvements. I recommend checking the article for a native speaker.

5. In conclusion, the authors should describe in more detail what their further research will focus on.

Author Response

Response to Reviewer 2 Comments

Point 1: Line 125 - Fig. 1 Misspellings: The lactation of Gonghe PV Power Plant in Qinghai province

Response 1: The right word is "location" We are very sorry for our negligence regarding this mistake. We have revised it.

Point 2: The data and descriptions in all graphs should be written in a larger font. In this form it is quite difficult to read them.

Response 2: Thank you for your suggestion .We have revised all graphs.

Point 3:  Line 203-204 Fig. 4 - Authors should use different colors or patterns to differentiate between curves. It is difficult to distinguish them in this form.

Response 3:Thank you for your suggestion .We have revised Fig. 2 and Fig. 4.

Point 4: The paper is well written, but it still needs some english improvements. I recommend checking the article for a native speaker.

Response 4: Thank you for your suggestion. We have asked the professional language editing service company American Journal Experts to polish the manuscript.

Point 5: In conclusion, the authors should describe in more detail what their further research will focus on.

Response 5: Thank you for your suggestion. We have rewritten the conclusion and supplemented future research. The details are as follows: “In the future, more field measurements at different heights should be observed and used to calibrate the results of climate models and satellite observations.” (Lines 382-384, Page 14)

The authors thank you very much again for your constructive comments which play a significant role to increase the quality of this paper.

Reviewer 3 Report

This paper presents a study about the foundation for research on the impact of large-scale PV development in deserts on the local ecological environment. For this propose, multi-point field observation data from January 2019 to December 2019, the impact of a large-scale PV development in the desert on the local air temperature and relative humidity was studied, and the characteristics and variations in the air temperature and relative humidity were evaluated on different time scales. Although this paper contributes to some novel knowledge to the related field, it needs some improvements before acceptance for publication. The major comments are as follows:

(1) This manuscript was submitted to Sustainability to be considered for publication but there are no references to this journal in the bibliography. Why did the authors choose this journal? Please add some references to show the relation of your work with the journal scope and compare (if possible) your results with those reported in the new references.

(2) Introduction: On december 2019, it took place the COP25 (United Nations Climate Change Conference) in Spain. Some conclusions can be included in the manuscript about environmental pollution and climate change as part of the introduction section of the paper.

(3) Line 53: Can you give some data about the increasing of the installed capacity of PV in the last years?

(4) Line 58: a reference is needed.

(5) Line 70: can you give the specific temperature increase? (As you do in the other cases).

(6) Lines 155-170: have you found any other studies (as such you mention in the introduction) that can corroborate your explanations?

(7) Line 206: you say "The change trend of relative humidity was exactly the opposite of the air temperature trend". But it is not true in most cases. For example, can you explain why in January, the maximum of relative humidity difference is not in relation with a minimum in the temperature? In May, June and July the same discordance can be found.

(8) Line 208: Was this an expected result? If yes, this observation must be pointed and some references must be used here.

(9) Lines 230-248: all your explanations are based on difrerent trends observed on the humidification effect. Can you compare your results with other previously published works with a similar effect? (in different seasons, at day, at night...).

(10) Some sentences that you have included in Results section must be moved to Discussion section because they explain the different observed effects reasons. In my opinion, these sentences may be moved to the Discussion section, where all the results must be justified.

(11) Lines 286 and 332: can you corroborate your justification by some references?

(12) Lines 340-344: these are very interesting sentences, but they are not part of your research. Therefore, I suggest to move these sentences to the introduction in order to reinforce there the importance of your study.

(13) Conclusions: some sentences are not directly conclusions of your work. Please remove them.

(Final): Author contribution, funding and conflicts of interest (all mandatory) are missed.

As a general comment, the manuscript must be adapted to the journal format for following submissions.

Author Response

Response to Reviewer 3 Comments

Point 1: This manuscript was submitted to Sustainability to be considered for publication but there are no references to this journal in the bibliography. Why did the authors choose this journal? Please add some references to show the relation of your work with the journal scope and compare (if possible) your results with those reported in the new references.

Response 1: Thank you for your suggestion. We have supplemented reference related to this manuscript in this journal. Unfortunately, the content of this manuscript cannot be compared with the articles in the journal. (Lines 63-64, Page 2)

Point 2: Introduction: On december 2019, it took place the COP25 (United Nations Climate Change Conference) in Spain. Some conclusions can be included in the manuscript about environmental pollution and climate change as part of the introduction section of the paper.

Response 2: Thank you for your suggestion. We supplemented some of the conclusions of this conference on greenhouse gas emissions and the direction of renewable energy development. (Lines 51-55, Page 2)

Point 3:  Line 53: Can you give some data about the increasing of the installed capacity of PV in the last years?

Response 3: We have added the data about the increasing of the installed capacity of PV in 2018 and 2019. (Lines 56-58, Page 2)

Point 4: Line 58: a reference is needed.

Response 4: We have added a reference here, which is “Research of solar energy development of Qinghai Province”, published in 2011. (Lines 69, Page 3)

Point 5: Line 70: can you give the specific temperature increase? (As you do in the other cases).

Response 5: We have supplemented the temperature increase value of large-scale PV power plants in the Sahara, which is the maximum temperature increase 1.28 K and the minimum temperature increase 0.97 K. (Lines 82, Page 3)

Point 6:Lines 155-170: have you found any other studies (as such you mention in the introduction) that can corroborate your explanations?

Response 6: We put the explanation in Section 4.1. The conclusion of the temperature change in our study was compared with Golmud and Tucson PV power plants, and explained the reasons different from those in these studies.

Point 7: Line 206: you say "The change trend of relative humidity was exactly the opposite of the air temperature trend". But it is not true in most cases. For example, can you explain why in January, the maximum of relative humidity difference is not in relation with a minimum in the temperature? In May, June and July the same discordance can be found.

Response 7: We are very sorry for our unclear expression in the paper. We have rewritten this sentence. The new text is as follows: “When the temperature was the highest at around 16:30 (the lowest at around 8:00), the relative humidity was the lowest (highest).” (Lines 237-238, Page 9)

We hope that this change made this sentence easier to understand.

Point 8: Line 208: Was this an expected result? If yes, this observation must be pointed and some references must be used here.

Response 8: This result is consistent with the conclusion of Golmud PV power plant and we supplement the reference. (Lines 239-240, Page 9)

Point 9: Lines 230-248: all your explanations are based on difrerent trends observed on the humidification effect. Can you compare your results with other previously published works with a similar effect? (in different seasons, at day, at night...).

Response 9: There are very few results on the relative humidity of the PV power plant. At present, the changes of relative humidity have not been analyzed from the seasonal scale, only the changes of daytime and night-time. We compared the daytime and night-time relative humidity of Golmud PV power plant at section 4.2. In the future, we will continue to follow and collect relevant information. (Lines 364-365, Page 13)

Point 10:  Some sentences that you have included in Results section must be moved to Discussion section because they explain the different observed effects reasons. In my opinion, these sentences may be moved to the Discussion section, where all the results must be justified.

Response 10: Thank you for your suggestion. We have adjusted part of the results and analysis to the conclusion.

Point 11: Lines 286 and 332: can you corroborate your justification by some references?

 Response 11: Thank you for your suggestion. We have supplemented some references.

Point 12:  Lines 340-344: these are very interesting sentences, but they are not part of your research. Therefore, I suggest to move these sentences to the introduction in order to reinforce there the importance of your study.

Response 12: Thank you for your suggestion. We have adjusted this part to the introduction and revised the introduction. (Lines 99-113, Page 3-4)

Point 13: Conclusions: some sentences are not directly conclusions of your work. Please remove them.

Response 13: This is a very good suggestion. Thank you very much. We have rewritten the conclusions.

(Final): Author contribution, funding and conflicts of interest (all mandatory) are missed.

As a general comment, the manuscript must be adapted to the journal format for following submissions.

Response: Thank you for your suggestion. We supplemented Author contribution, Funding and Conflicts of interest.

The authors thank you very much again for your constructive comments which play a significant role to increase the quality of this paper.

Round 2

Reviewer 1 Report

All the comments have been answered with corresponding revisions. I suggest it could be accepted.

Reviewer 3 Report

Since the authors have improved their manuscript in terms suggested by this reviewer, I recommend to publish it in its actual form.